# New Tumor Budding Evaluation in Head and Neck Squamous Cell Carcinomas

**DOI:** 10.3390/cancers16030587

**Published:** 2024-01-30

**Authors:** Claudio Cacchi, Henrike J. Fischer, Kai Wermker, Ashkan Rashad, Danny D. Jonigk, Frank Hölzle, Maurice Klein

**Affiliations:** 1Institute of Pathology, School of Medicine, University Hospital RWTH Aachen, Pauwelsstrasse 30, 52074 Aachen, Germany; ccacchi@ukaachen.de (C.C.); djonigk@ukaachen.de (D.D.J.); 2Institute of Immunology, School of Medicine, University Hospital RWTH Aachen, Pauwelsstrasse 30, 52074 Aachen, Germany; hefischer@ukaachen.de; 3Department of Oral and Cranio-Maxillofacial Surgery, Klinikum Osnabrück GmbH, Am Finkenhügel 1, 49076 Osnabrück, Germany; kai.wermker@klinikum-os.de; 4Department of Oral and Maxillofacial Surgery, School of Medicine, University Hospital RWTH Aachen, Pauwelsstrasse 30, 52074 Aachen, Germany; arashad@ukaachen.de (A.R.); fhoelzle@ukaachen.de (F.H.); 5German Center for Lung Research (DZL), Breath Hanover, 30625 Hanover, Germany

**Keywords:** tumor budding, oral cancer, HNSCC, cancer, tumor marker, tumor

## Abstract

**Simple Summary:**

There have been no analyses of tumor budding (TB) in different margin sections and using the fixation method in HNSCC in the literature. The mean TB (TB rel) of all tumor-positive margin sections (n = 443) of the primary tumor was analyzed in an FFPE-fixed tumor from 66 patients with HNSCC, and they were compared with cryo-fixed sections. TB rel correlates significantly with tumor aggressiveness. TB often varies between the different tumor margins of FFPE sections of the same patient, and they differ depending on the fixation method. Our data show that a randomly selected margin section does not reliably reflect the TB, and thus, it cannot predict the prognostic outcome. TB rel could compensate for the differences in TB score analysis. The determination of the TB score in cryo sections seems to be inaccurate compared with analyses in FFPE. The method shown here is cost effective, easy to integrate into a clinical workflow, and seems useful for future studies.

**Abstract:**

Background: Tumor budding (TB) is a histomorphological characteristic of the tumor invasion front and it has an impact on the tumor outcome prediction for head and neck squamous cell carcinoma (HNSCC) aetiopathology. Patients and methods: The average TB score (TB rel) of all tumor-positive marginal sections (n = 443) in the primary tumor was analyzed in the FFPE-fixed tumor slices of 66 patients with HNSCC, and they were compared with cryo-fixed sections. Results: TB rel correlates with tumor aggressiveness (i.e., lymph node metastasis quantity, lymph node ratio, extra capsular growth, Pn1, pV1, grading). The TB scores often vary between the different tumor margins of FFPE sections in the same patient, and in many cases, they differ depending on the fixation method. Conclusion: Our data show that a randomly selected marginal cut cannot reliably mirror the TB score, and thus, they cannot predict the prognostic outcome. However, TB rel could be a tool that compensates for differences in TB score analysis. TB score determination in cryo sections seems to be inaccurate compared with TB determination in FFPE.

## 1. Introduction

The most frequently found tumor in the head and neck (>90%) is squamous cell carcinoma (HNSCC). The incidence of HNSCC in Europe is approx. 140,000 cases/year. This corresponds to a share of 4% of all malignancies in adults [1]. The prognosis is poor, with a 5-year survival rate of <50%, and a high recurrence rate of 15–40%, over the course of time [2].

A preliminary selection of patients with lymph node metastases with the help of preoperative prepared biopsy and histological reconditioning could prevent an overtreatment of non-indicated neck dissection. A potential candidate for risk classification could be the tumor budding (TB) score, which is evaluated via hematoxylin-eosin staining (HE). TB is defined as an accumulation of isolated cells or cell clusters (<5 cells) that are separated from the tumor invasion front (TIF) [3].

TB has already been investigated in many tumors (colon carcinoma, gastric cancer, urothelial carcinoma, pancreatic carcinoma, breast cancer, endometrial carcinoma, lung cancer, esophageal cancer, liver cancer) [4,5,6,7,8,9,10,11,12]. In many tumors, the TB score is associated with a worse prognosis [13,14,15,16,17,18]. There is also evidence to suggest that TB in HNSCC has a predictive benefit, with regard to different tumor localizations [19,20,21,22,23,24,25].

Almangush et al. showed in a meta-analysis that the TB in oral squamous cell carcinomas (OSCC) is correlated with a high risk for lymph node metastasis (LNM), a reduced disease specific survival rate, and a reduced overall survival rate [26].

Xie et al. state that the TB evaluation is an easy and well reproducible method, and it should be incorporated into every pathologic report. The team could show in 255 OSCC patients that an increased TB score is an independent prognostic factor. They found a significant correlation between TB, differentiation (*p* = 0.048), LNM (*p* = 0 < 0.001), and invasive pattern (*p* = 0 < 0.001) [27]. The literature search for HNSCC revealed that the tissue used for the analysis of TB is always HE-stained, formalin-fixed, and paraffin-embedded tissue (FFPE) [28,29]. However, up until now, no analyses have been performed on cryo-fixed tissue. In addition, our aim was to evaluate if the assessment of TB and its potential role as prognostic indicator differs between FFPE vs. cryo-fixed HNSCC tissue.

In summary, TB is a significant prognostic morphological tumor marker. The primary aim of this study was to analyze all the tumor-positive histological margins of the tumor infiltration front in surgically-resected HNSCC to characterize TB as a prognostic marker in HNSCC. More specifically, we tried to answer the following questions: Are TB characteristics reliably identified in several slices of the same tumor? Could this have an impact on risk stratification or treatment decisions?

## 2. Material and Methods

### 2.1. Ethics Statement

This study was approved by the local ethics committee (Ethical Committee of the Universitätsklinik RWTH Aachen, Approval-No. EK 22-373) and it was conducted in accordance with the Guidelines for Good Clinical Practice, and in compliance with the Declaration of Helsinki. All patients gave their written informed consent to participate in this study.

### 2.2. Patients and Clinical Data

Patients with a histologically proven HNSCC and an age > 18 years were included. Patients with incomplete data sets for the post-operative staging of the histologic specimens were excluded from the study. Patients with tumor-recurrence or with a second HNSCC were also included.

Every patient underwent preoperative staging; at minimum, a head and neck CT, MRT, X-ray thorax, and abdominal sonography were performed. The follow-up data and tumor characteristics were collected retrospectively from the database hospital information system.

The lymph node ratio (LNR) was also determined (lymph node metastasis: all lymph nodes in neck dissection = LNR).

### 2.3. Tissue Samples and Histochemistry

Patients classified before 2017 with a 7th TNM UICC classification were reclassified in accordance with the current 8th TNM UICC Classification. All analyses, and the whole histological staining process, were performed at the Department of Pathology and in the centralized-Biomaterial-Bank RWTH-University Aachen, Germany. The HNSCC tumor tissue was collected by the Department of Oral- and Maxillofacial surgery of the University of Aachen.

Once the patient collective was established, the FFPE sections of the respective primary tumor that was associated with the patient were made available for TB evaluation.

The FFPE slices’ thickness was 4 µm, and they were stained with HE (Eosin solution: Eosin G solution 1% aqueous; Haematoxylin: Haematoxylin M (modified in accordance with Mayer). The staining was conducted on the Tissue-Tek Prisma staining machine from Sakura (Umkirch, Germany).

To approach the secondary aim of this study, and to investigate whether the assessment of TB and its potential role as a prognostic indicator differs between FFPE and cryo-fixed HNSCC tissue, 4 µm cryo slices were prepared and stained with HE. The cryo slides were selected in a standard operating procedure after tumor resectioning occurred. Tissue samples (frozen aliquods) were taken from the area of the primary tumor, to obtain the greatest amount of tumor tissue. Every aliquod was tested for suitability by a pathologist.

### 2.4. Analysing of Tumorbudding in FFPE Slices

Two investigators (MK and CC) analyzed TB and came to a shared decision; they were unaware of the clinical data beforehand. The evaluation of the 4 µm stained slices was performed using an Olympus BX51 microscope (Hamburg, Germany); pictures of slices were taken with the Olympus EP 50 camera (Münster, Germany).

After the identification of a representative hot spot in the tumor-invasion-front (TIF), using 10× magnification, the TB score was determined in this hot spot using a magnification of 20×. For the FFPE samples, all tumor-positive slides were analyzed. If step cuts were applied in a circular cut position, the highest TB scores were selected for this circular cut position. Moreover, the tumor-slides’ position, taking bone tissue and previous decalcifications into consideration, were included.

A TB was defined as a single tumor cell, or a tumor cell cluster of 4 or fewer tumor cells [30]. The TB was interpreted as follows: TB score 1 (0–4 buds = low TB), TB score 2 (5–9 buds = medium TB), and TB score 3 (>10 buds = high TB). The analysis of the TB score was conducted in accordance with the study by Xie et al. [27]. For each tumor tissue slide, TB was analyzed in order to differentiate between intra- and peritumoral tumor budding (ITB; PTB). To investigate whether TB features are reliably detectable in multiple slides of the same tumor, all positive marginal sections must be analyzed. For the FFPE samples, the average TB score of all tumor-positive slides was defined as TB relative (sum of TB of single slide/all tumor positive tumor slides = TB rel).

Figure 1 shows tumor budding in a FFPE-fixed tissue slide (HE, magnification 20×). The high-power field (HPF) presented a low TB, and thus, a TB score of 1.

### 2.5. Analysing of Tumorbudding in Cryo Fixed Slices

The TB of the single cryo HE section was analyzed, as described above. The TB score and data acquisition of cryo samples was performed blind on the source of the FFPE samples; the ITB and PTB were also determined. We first provided a general overview of the TB analysis, which was conducted on the cryo slide, and second, we provided a comparison with the FFPE slides.

Figure 2 shows TB in a cryo-fixed tissue slide (HE, magnification 20×). The HPF presents a low TB and a TB score of 1.

### 2.6. Statistical Analysis

The data were tested for normality using the Shapiro–Wilk test. No analyzed data set was normally distributed. Comparisons between two groups were thus calculated using the Mann–Whitney test; correlations were identified by calculating the Spearman r. * *p* < 0.05, ** *p* < 0.01, ns = not significant. The program, Excel 2016 for Windows, by Microsoft (Redmond, WA, USA), was used for descriptive analyses (patient characteristics). The statistical analyses were performed using GraphPap PRISM Software (San Diego, CA, USA) version 9.5 for Windows. Correlations between the TB, tumor characteristics, tumor stadium (UICC), disease progression, recurrence, lymph nodes or distant metastases, and capsule breakthrough were analyzed. The TB rel was correlated with the clinical and pathologic outcome parameters and LNM, in addition to controlling suitability regarding patient selection. In each patient, the result of the cryo slide’s TB was then compared with the TB rel of the FFPE samples. The TB rel was then compared with the single cryo slide’s TB.

## 3. Results

### 3.1. Patient Characteristics

Sixty-six patients (30 female; 36 male) were included, with a mean follow-up time of 18.9 months. The age at first diagnosis was 66.2 years on average. The included patients had different tumor localizations (floor of mouth, 16 (24.2%), tongue, 16 (24.2%), cheek (mucosa), 6 (9.1%), lower jaw, 15 (22.7%), upper jaw, 8 (12.1%), oropharynx, 5 (7.6%)), and the tumors differed in terms of severity (unilateral n = 58; 87.9%, bilateral n = 8; 12.1%).

Clinical pathological values, TNM data, and prognostic data were collected, as follows: Grading (G1 n = 2; 3%, G2 n = 50; 75.8%, G3 n = 14; 21.2%), TNM: T (pT1 n = 7; 10.6%, pT2 n = 22; 33.3%, pT3 n = 15; 22.7%, pT4a n = 20; 30.3%, pT4b n = 2; 3%), N (pN0 n = 34; 51.5%, pN1 n = 7; 10.6%, pN2a n = 3; 4.5%, pN2b n = 10; 15.2%, pN2c n = 5; 7.6%, pN3 n = 7; 10.6%), M (M0 n = 66; 100%), blood vessel invasion (pV; no n = 62; 93.9%, yes n = 4; 6.1%), perineural invasion (Pn; no n = 34; 51.5%, yes n = 32; 48.5%), lymph vessel invasion (pL; no n = 58; 87.9%, yes n = 8; 12.1%), extracapsular spread (no n = 55; 83.3%; yes n = 11; 16.7%), resection status (R; R0 n = 61; 92.4%, R1 n = 5; 7.6%), UICC stage (UICC stage I n = 8; 12.1%, UICC stage II n = 11; 16.7%, UICC stage III n = 9; 13.6%, UICC stage IVa n = 29; 43.9%, UICC stage IVb n = 9; 13.6%), recurrent tumors (No n = 55; 83.3%, yes n = 11; 16.7%), and ASA score (American Association of Anesthesiologists; ASA 1 n = 4; 6.1%, ASA 2 n = 32; 48.5%, ASA 3 n = 29; 43.9%, ASA4 n = 1; 1.5%).

### 3.2. Analysis of TB in FFPE Slides

To determine whether differences exist between multiple positive margin sections of an FFPE tumor material, all tumor-positive margin sections were subjected to TB analysis. In addition, the exact localization of TB was determined. For proof of suitability of TB rel, a comprehensive analysis of every slide was performed.

Four-hundred and forty-three FFPE slides were analyzed in total. Most slices (n = 429; 96.84%) were analyzed in the peritumoral environment, and the rest of the slices (n = 14; 3.16%) only showed intratumoral TB (ITB). The distribution of the TB score was as follows: TB1 (n = 314; 70.9%), TB2 (n = 104; 23.5%), and TB3 (n = 25; 5.6%).

Furthermore, the differences in TB between the positive margin FFPE slides were analyzed for each patient. In 36 cases, we found deviations between the slices with one TB score point. In seven cases, there was a difference of two TB score points. Due to the differences between the TB scores of the single sections of the tumor, the correlations between the clinical pathological parameters were ascertained using the TB rel.

### 3.3. Increased TB Rel Correlates with the Number of Lymph Node Metastases, the Lymph Node Ratio, and Extracapsular Growth

To test the predictive utility of TB rel regarding the risk of lymph node metastasis, TB rel needed to correlate with various aggressiveness characteristics of LNM (LNM quantity, LNM/lymph node = lymph node ratio (LNR), pN status, LNM with capsular spread).

Our data indicate that TB rel was not significantly correlated with pN status (Figure 3A). Another aggressiveness characteristic of LNM is capsule breakthrough. Here, our data show a significant difference between increased TB rel and extracapsular growth (*p* = 0.01; Figure 3B). Nevertheless, TB rel correlated with the amount of LNM (*p* < 0.001; Figure 3C) and LNR (*p* < 0.001; Figure 3D) to a significant extent.

### 3.4. Increased TB Rel Is Correlated with Decreased Lymph Node Metastasis Free Time

In addition, the patient population was examined for the presence of LNM in the follow-up. There was a significant difference in terms of the risk of developing LNM during follow-up, with a TB rel > 2 (TB rel 1 vs. TB rel > 2; *p* = 0.018; n = 5; Figure 4). Figure 4 shows a Kaplan–Meyer curve that is dependent on the TB rel value and the LNM-free time. There is a significant trend (*p* = 0.0329) concerning the TB rel score classification shown here.

### 3.5. TB Rel Significantly Correlates with Perineural Status, Bloodvessel Invasion, and Less Tumor Differentiation

An analysis was also established to explore TB rel and other clinicopathologic parameters. A higher TB rel was correlated with Pn status (*p* < 0.05) and blood vessel invasion (*p* < 0.01). Grading showed a significant difference between G2 and G3 with TB rel (*p* < 0.05; Figure 5A,B,F). In contrast, there was no significant difference between the presence of lymphatic vessel invasion, distant metastases (DM), or local relapse with TB rel (Figure 5C–E).

### 3.6. Trend of Increased TB Rel Correlates with Worse Overall Survival

In addition, an analysis was performed for TB rel cut-off values (1, 1–1.5, 1.5–<2 and >2), the overall survival rate (Figure 6A), and the disease-specific survival rate (Figure 6B). The Log-rank test showed a significant trend of *p* = 0.0329 for the overall survival rate, and a non-significant trend of *p* = 0.5763 for all cut-off values concerning the disease-specific survival rate. In addition, there was a significant difference (*p* = 0.0180) between overall survival rates (Figure 6A) due to the differences between TB rel 1 vs. TB rel > 2 in the Log-rank test.

### 3.7. Poor Match concerning Tumor Budding between FFPE and the Cryo-Fixed Tumor Tissue of the Same Patient

The secondary aim of this study was to investigate whether TB, and its potential role as a prognostic indicator, behaves differently in FFPE compared with cryo-fixed HNSCC tissue; a comparison was made between TB rel (from FFPE slides) and TB (in one cryo slide) in the same patient. In the following section, we first provide a general overview of the TB analysis of the cryo slides, and second, we show the comparison with the FFPE slides.

Intra- and peritumoral TB was determined in all cryo slides ((n = 66): ITB (n = 37; 56%), PTB (n = 29; 44%)). The distribution of the TB score levels was as follows: TB 1 (n = 53; 80.3%), TB 2 (n = 12; 18.2%), TB 3 (n = 1; 1.5%).

To verify whether the cryo slides could be used for diagnostic purposes, they were compared with the TB, in FFPE sections, as a standard analytical method. TB rel was used to compensate for the differences in different sections of the same tumor. The TB rel was compared with one cryo section of the same patient. When the TB score of the cryo slides was compared with the highest TB score (highest TB score of a single section of a patient) of the FFPE slices, there was a difference between TB scores in 34 cases.

For the graphical representation of this divergence, the FFPE slides, TB rel and cryo slides, and TB were directly compared (Figure 7A). There is a clear deviation between the TB score of the TB rel (FFPE) and TB (cryo). Figure 7B–D shows the graphical representation of each TB score/number for all tumor positive margin slides that are in agreement with/which diverge from the respective TB scores of the cryo-collective of the same patient. Again, there is poor agreement between the analysis of TB in the FFPE slides and TB in the cryo slides.

### 3.8. Summary of Results

TB rel correlates with tumor aggressiveness (LNM quantity, LNR, extracapsular growth, Pn1, pV1). In addition, there also appear to be correlations between increased TB rel, the risk of LNM during the follow-up appointment, and a decreased overall survival rate. TB scores often differ in the different tumor margins of the FFPE sections of the same patient. TB scores often differ depending on the fixation method chosen.

## 4. Discussion

### 4.1. Strengths and Weaknesses of the Study

Our study was the first to evaluate the prognostic potential of TB compared with TB rel in FFPE and cryo-fixed tissue in HNSCC; we aimed to find the best analysis method and scoring system. Our data show that the prognostic evaluation of TB in HNSCC is dependent on both the tissue fixation method and the slices analyzed. However, there are also limitations of this study, as follows: the retrospective type of study and small sample size. In contrast, a strength of this study is the heterogenic localization of head and neck squamous cell carcinomas. This reflects the realistic, every day, clinical situations of different primary localizations, and the different levels of disease severity. A particular strength of our study was that for each individual patient, a comparison between FFPE and cryo tissue sectioning was performed.

In addition to the clinical applicability of the TB rel, correlations between known risk factors also served to check whether the collective is suitable for analysis. As TB rel increases, tumor aggressiveness increases, as shown via known prognostic factors (LNM, lymph node quotient, LNM capsular growth).

Another weakness of the study is that only a single slide was used for the cryo tissue slide. In addition, it must be questioned whether TB scoring would have been more accurate in the cryo sections if more than one slice had been analyzed. Depending on this, the agreement with the TB rel of the FFPE sections could also have changed.

Nevertheless, we aimed for a standardized study design. To improve the design, each tissue would have to be embedded, first as cryo, then as FFPE, and then they would need to be compared against each other. This is not possible as a retrospective study design and is also ethically critical in terms of the diagnostic tumor work-up.

### 4.2. TB and Staining

Togni et al. discuss the need for a standardized scoring system for HNSCC [31]. In our study, only HE was investigated as a staining method. However, other staining methods, especially immunohistochemical staining methods, were analyzed in the literature for TB determination.

During the manual detection of TB, hematoxylin and eosin staining, or pancytokeratin-immunostained sections, were mainly used [31]. Abd Raboh showed, in 118 cases of laryngeal cancer, that there are differences between recurrence and distant metastasis free survival when HE and pancytokeratin-immunostaining were compared. They recommended inclusion of TB in every histopathology report due to the prognostic utility [32]. This might be because there are beneficial differences between TB in cryo vs. FFPE tissue when it is stained with panzytoceratin vs. HE.

This could be important because of its potential application in fast cutting analyses, and the standard of the quicker and cheaper HE compared with immunohistochemical staining methods.

However, in a review article by Kale and Angadi, the authors recommend the HE assessment for a comparison of cytokeratin immunochemistry when analyzing TB [33]. HE staining is low cost and efficient, and it makes a global evaluation of TB possible [30].

### 4.3. Suitability of TB Rel as a Tumor Marker Compared with TB in a Single Slide

The primary aim of this study was to analyze all positive margin sections of the TIF of the primary tumor, and to determine whether TB rel is a reliable factor for risk stratification compared with TB analysis in a single slice.

More specifically, we tried to answer the following questions: Are TB characteristics reliably identified in several slides of the same tumor? Could this have an impact on risk stratification or treatment decisions? The consensus is that TB is useful for prognostic valuations [33]. Typically, a random tumor margin on an FFPE tissue slide is used for the analysis of TB. This study was the first to analyze all tumor-positive histological slides of the primary tumor in HNSCC. Our data show that a randomly selected edge cut can vary during the evaluation, and thus, it can consecutively change the prognostic outcome. Our data suggest that diagnostic or therapeutic decision-making is dependent on the choice of the slide. Thus, for a final validation of the prognostic properties of the TB score, larger cohorts need to be analyzed.

It appears that the risk of an unfavorable LNM constellation (N+, extracapsulary growth of LNM, LNM in follow up) begins at a TB rel ≥ 2. This may be due to a reduced TB rel analysis (TB rel > 2; TB rel < 2) with regard to the LNM assessment; subsequent studies could simplify the evaluation.

Talmi et al. showed in a review that the LNR (LNM: all lymph nodes in neck dissection = LNR) might be a useful and promising prognostic marker [34]. The correlation between TB rel and LNR shows that with increased TB, the risk of LNM is increased. It seems interesting for future studies whether TB rel can be used to identify the clinically false negative cN0 neck (cN0 but pN+ neck).

Furthermore it could be interesting to analyze the TB rel in the LNM tissue and ascertain whether there is prognostic relevance; for instance, studying the morphology of LNM in malignant melanoma cases [35]. A comparison of TB rel in the primary tumor comprising TB in LNM also seems like a possible future research direction; here, it might be possible to ascertain whether the same TB could be found. This could contribute to the understanding of metastasis to the lymph nodes. In addition, it could be investigated which histomorphological correlate is prognostically useful or decisive for LNM.

### 4.4. Effect of the Results Analyzing ITB and PTB

The exact description of localization concerning TB in the tumor PTB or ITB is important to enable comparability. Some authors analyzed the prognostic effect of ITB and PTB. Lugli et al. note that the ITB is a promising biomarker for the preoperative management of colorectal cancer, but further research is needed to recommend an analysis of ITB [30]. Interestingly, the analysis of ITB was much more difficult in our study than PTB. The spaces between tumor nests were sometimes very small, so the identification of TB usually resulted in lower TB scores.

The FFPE material showed more PTB because of the resectioning of tumor with a safety margin. The cryo slices show different results because of the extraction of the tumor tissue via biobanking. We might speculate that studies which only analyze cryo slices find no differences in terms of TB, and they might have lower TB scores than FFPE slices. Slide scope scanners and computer aided evaluation methods could be more advantageous than manual evaluation methods.

It may also be possible to derive therapeutic benefits from the diagnosis of IPT and TB. The analysis of ITB in rectal cancer was able to identify those tumors which respond poorly to neoadjuvant chemoradiotherapy and have a poor prognosis [36].

However, a comparison of TB values from the tumor center with the tumor margin cut could differ. Future studies should also investigate this.

### 4.5. TB Analysis in Cryo-Fixed Tissue Does Not Correlate with FFPE Tissue and Is Not Suitable for Prognostic Assessments

The secondary aim of this study was to evaluate if the assessment of TB and its potential role as prognostic indicator differs between FFPE and cryo-fixed HNSCC tissue. The focus was to compare the differences between analyses, the prognostic impact of cryo slices, fast-cut similar techniques, and classic HE FFPE tissue analyses in TB, rather than the prognostic impact of different tumor localizations.

Generally, the microscopic assessment of FFPE slides was easier to analyze compared with cryo-fixed tissues because the artefacts of frozen cuts, and the inconsistent quality resulted in a poorer assessment of TB in the cryo tumor tissue.

This had the effect of the single cells or the cell nest being more difficult to identify than in FFPE. The study showed that there was heterogeneity in both methods.

During the study design process, we proposed that there could be an intraoperative use of TB; for example, more radical resectioning in parts where there was a higher TB score could take place. This prompted the following question: is there a clinical use for TB as part of an intraoperative frozen fast cut technique? To answer this question, the secondary aim of this study was to evaluate if the prognostic use of TB differs between FFPE HNSCC tissue slices and cryo HNSCC tissue slices.

This study supports the hypothesis that the intraoperative use of TB as part of a fast cut technique is difficult or impossible. It is possible that special techniques involving immunohistochemistry with epithelial markers (e.g., cytokeratin) could solve the problem in the future.

### 4.6. Clinical Application and Effect Depending on the Tissue Material Used

In addition to the prognostic and diagnostic use of TB, a therapeutic application is also conceivable. In a systematic review of Almangush et al., it was shown that preoperative TB is important for prognostic outcomes, LNM, as well as disease-specific and overall survival. The authors postulate that there is perhaps a use for TB scores in the evaluation of tumor aggressiveness, and this could improve therapeutic decisions [13]. For therapeutic applications, the determination of the fixation and evaluation methods must be discussed.

Based on the data available here, FFPE materials should be recommended as part of diagnostic fixation methods. Furthermore, TB rel compensates for outliers at all extremes, and it may be better suited for TB determination in the HE slide. This may also provide greater reproducibility with regard to therapeutic decision-making.

A threshold for TB rel may be needed to reduce bias in different TB results, and possibly in decision-making processes after analysis.

In tongue squamous cell carcinoma, TB correlates with survival and LNM [37]. TB is an important risk factor for predicting LNM in all stages of OSCC, and it is associated with poorer outcomes in early stage tumors [38]. The TB rel in this study correlates with LNM, lymph node quotient, and LNM capsular growth. This could have an influence on surgery; for example, more adjustments to the surgical process of neck dissections in parts of TB could occur. For a preoperative TB determination, FFPE tissue slices should be used as it results in better TB detection.

It is possible that the TB could get an impact of indication of radio therapy like the perineural invasion (Pn1). In this context, the TB rel might be the better method for analyzing TB and balancing expression variations.

### 4.7. Outlook

The accurate analysis of TB is important. This study was able to show that a possible future therapeutic decision may be influenced by the number of slices, and that the TB rel could potentially produce a more valid result than the TB analysis of a single slice. The material used for the preoperative risk stratification of lymph node metastasis should be FFPE because of the differences of cryo vs. FFPE.

These different morphologic analyses of TB in all tumor-positive histological specimens show tumor heterogeneity in TIF. This could explain the fact that many studies exhibit differences in terms of protein expression in TIF and prognostic outcome [39]. A combination of TB and tumor markers could further our understanding of more of the complex interactions between tumor environment, the morphologic aspects of the tumor, and its aggressiveness.

## 5. Conclusions

Moreover, TB rel could be a tool with which to compensate for differences in TB analyses of primary tumors. This cheap and useful marker has a high potential for further studies. The TB in cryo-fixed tissue slices is likely not usable for intraoperative analyses using fast-cutting, controlled surgery.

## Figures and Tables

**Figure 1 cancers-16-00587-f001:**
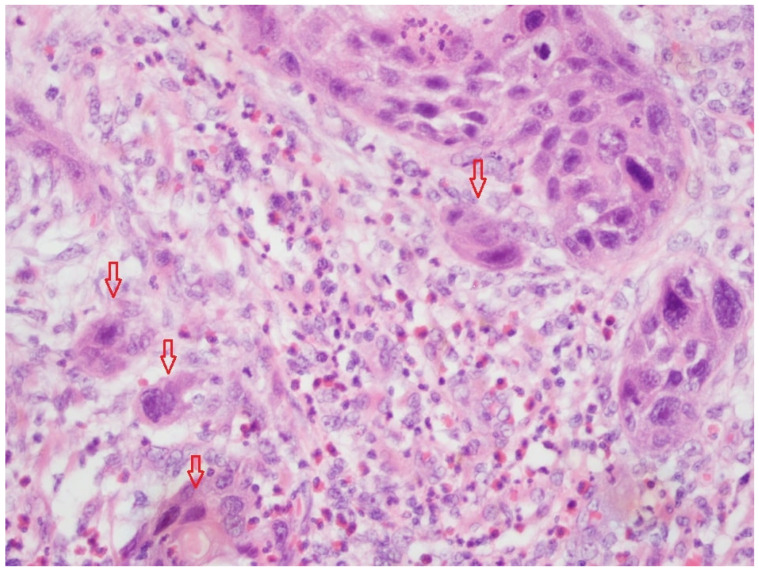
Example section of FFPE-fixed HNSCC tissue (HE, magnification 20×) with tumor budding. The tumor budding with a tumor cell nest (<4 cells; red arrows) is clearly visible on the left-hand side and upper right-hand side. This HPF presented low tumor budding (TB score 1).

**Figure 2 cancers-16-00587-f002:**
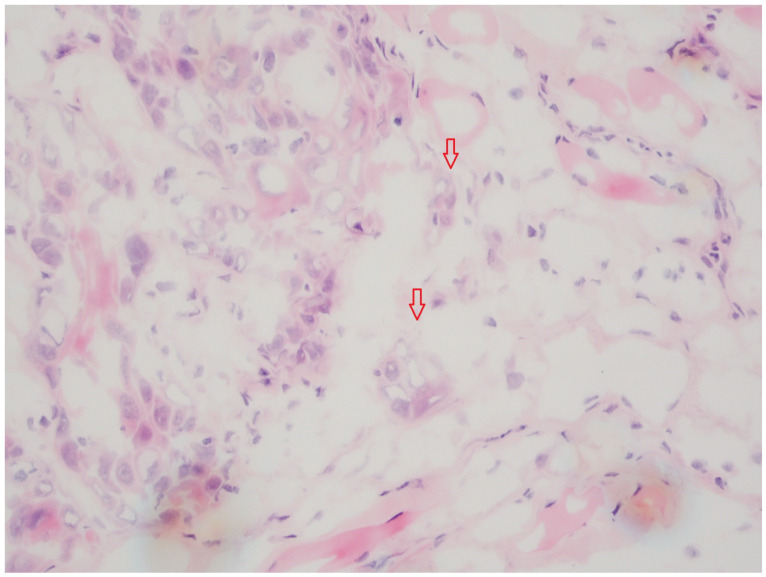
Example section of cryo-fixed HNSCC tissue (HE, magnification 20×) with tumor budding. The red arrows show the tumor budding nest of the cell (<4 cells/TB). This HPF presents low tumor budding (TB score 1). It is clearly visible that the tissue quality is poorer due to the cryo fixation process, compared with FFPE sections (Figure 1).

**Figure 3 cancers-16-00587-f003:**
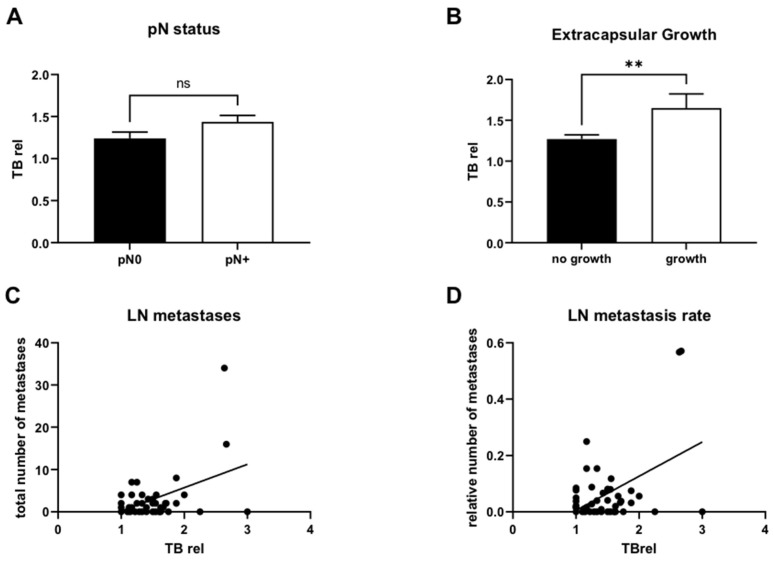
TB rel correlates with extracapsular growth, the number of lymph node metastases, and lymph node metastasis/lymph node quotient. Figure 3 shows in (**A**) that pN status (pN0 vs. pN+) does not correlate significantly with TB rel. In contrast, (**B**) shows that increased TB rel is significantly (*p* < 0.01) associated with the extracapsular growth of lymph node metastasis. (**C**) graphically shows how TB rel is associated with the total number of lymph nodes. As TB rel increases, the number of lymph node metastases increases. (**D**) shows that the relative lymph node metastasis (lymph node ratio = lymph node metastasis/lymphnodes) also increases as the TB rel increases. ** < 0.01, ns = non significant).

**Figure 4 cancers-16-00587-f004:**
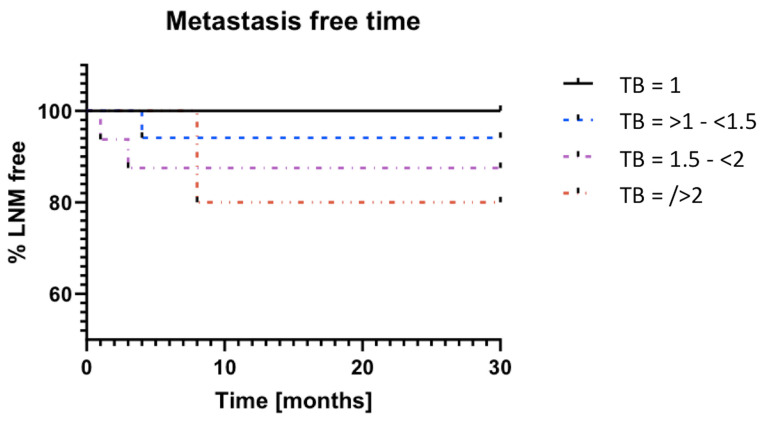
Increased TB rel is correlated with reduced lymph node metastasis free time. Figure 4 shows that the lymph node free time (during follow up) reduces as TB rel increases. There is a significant trend (*p* = 0.0329) shown here, regarding the TB rel score classification.

**Figure 5 cancers-16-00587-f005:**
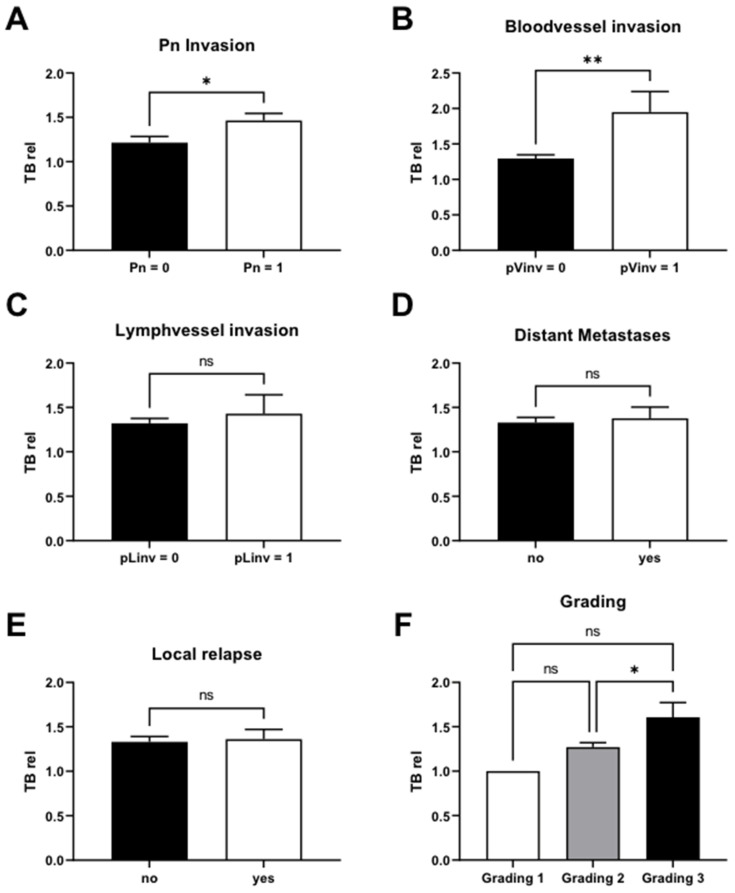
TB rel correlates significantly with perineural status, bloodvessel invasion, and less differentiation between tumors. A higher TB rel correlated with Pn status ((**A**); *p* < 0.05) and blood vessel invasion ((**B**); *p* < 0.01). In contrast, there was no significant difference between the presence of lymphatic vessel invasion, distant metastases (DM), or local relapse with TB rel (**C**–**E**). Regarding (**F**) the Grading, it showed a significant difference between G2 and G3 with TB rel (*p* < 0.05). (* < 0.05, ** < 0.01, ns = non significant).

**Figure 6 cancers-16-00587-f006:**
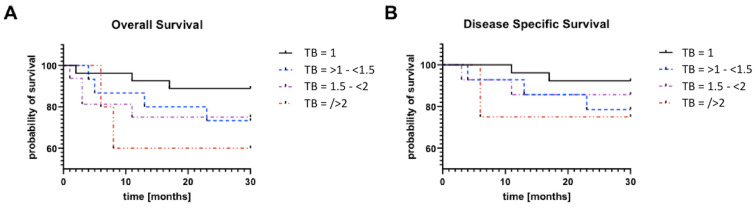
Trend concerning increased TB rel, which correlates with a worse overall survival rate. (**A**) shows TB rel vs. overall survival rate and (**B**) the TB rel vs. disease specific survival rate. The trend concerning the increased TB rel over all TB rel cut-offs (1, 1–1.5, 1.5–<2 and >2) was found to be significant for the overall survival rate ((**A**); *p* = 0.0329). This correlation could not be shown for the disease-specific survival rate (**B**).

**Figure 7 cancers-16-00587-f007:**
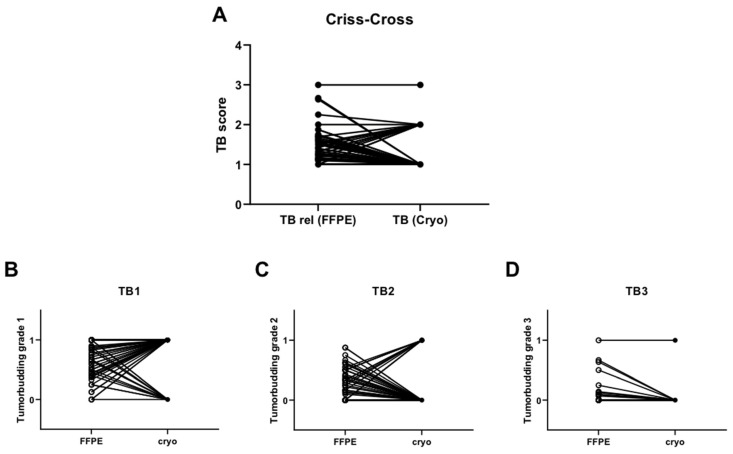
The TB consistency between the cryo- and FFPE-fixed slides of the same patient is poor. Figure 7 shows the comparison between TB analyses, which differed in terms of the fixation method. It can be clearly seen in (**A**) that TB rel in the FFPE section, and TB in the individual cryo-fixed section, diverge. This can also be seen in (**B**–**D**), regarding the TB score, concerning the total number of sections (TB score/number of the tumor-positive sections of the respective patient).

## Data Availability

The research data can be requested after consultation with the corresponding author.

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
