# Peer review of "New Tumor Budding Evaluation in Head and Neck Squamous Cell Carcinomas"

_cancers, 2024, doi:10.3390/cancers16030587_

Round 1

Reviewer 1 Report

Comments and Suggestions for Authors

Dear editor and authors, This is a well-prepared study which aim to evaluate New Tumor Budding in Head and Neck Squamous Cell Carcinomas . Their findings prove a randomly selected marginal cut cannot reliably mirror the TB score and thus cannot predict the prognostic outcome. However, TB rel could be a tool to compensate for differences in TB score analysis. TB score determination in cryo sections seems to be inaccurate compared to TB determination in the FFPE. It presents a well-delineated method and adequate processing of the data for a definite outcome.

Author Response

Thank you very much for these positive comments! We completely agree with you. The data shows that TB rel might be much better suited to perform TB scoring in HNSCC. Based on this data, FFPE seems to be the more suitable tissue compared to cryo-fixed tissue.

Reviewer 2 Report

Comments and Suggestions for Authors

No comment.

Author Response

Thank you very much for accepting the manuscript.

Reviewer 3 Report

Comments and Suggestions for Authors

The manuscript is well-written and describes work which is interesting and novel as well as having been executed to a high standard. The background of the subject is introduced well as is the work of others in the field. The methods are described comprehensively, as are the results with the statistical analysis being thorough. Particularly impressive is the depth of the discussion. I do not have any hesitation in recommending publication with the reservation that I am not a cancer cell biologist. 

Author Response

Thank you for your kind and friendly comment and recommendation for acceptance. Thank you for the comment on the discussion, a lot of effort has gone into this work.